ⓐ | Open Peer Review | *Clinical Microbiology* | Research Article

# The imbalance of pulmonary Th17/Treg cells in BALB/c suckling mice infected with respiratory syncytial virus-mediated intestinal immune damage and gut microbiota changes

Jiling Liu,[1,2] Yixuan Huang,[3] Nian Liu,[4] Huan Qiu,[5] Xiaoyan Zhang,[1] Xiaojie Liu,[1] Maozhang He,[1] Mingwei Chen,[3] Shenghai Huang[1,6]

**ABSTRACT**    The immune response induced by respiratory syncytial virus (RSV) infection is closely related to changes in the composition and function of gastrointestinal microorganisms. However, the specific mechanism remains unknown and the pulmonary-intestinal axis deserves further study. In this study, the mRNA levels of ROR-γt and Foxp3 in the lung and intestine increased first and then decreased. IL-17 and IL-22 reached the maximum on the third day after infection in the lung, and on the second day after infection in the small intestine and colon, respectively. Reg Ⅲγ in intestinal tissue reached the maximum on the third day after RSV infection. Moreover, the genus enriched in the RSV group was *Aggregatibacter*, and *Proteus* was reduced. RSV infection not only causes Th17/Treg cell imbalance in the lungs of mice but also leads to the release of excessive IL-22 from the lungs through blood circulation which binds to IL-22 receptors on the intestinal surface, inducing Reg Ⅲγ overexpression, impaired intestinal Th17/Treg development, and altered gut microbiota composition. Our research reveals a significant link between the pulmonary and intestinal axis after RSV infection.

**IMPORTANCE**    RSV is the most common pathogen causing acute lower respiratory tract infections in infants and young children, but the complex interactions between the immune system and gut microbiota induced by RSV infection still requires further research. In this study, it was suggested that RSV infection in 7-day-old BALB/c suckling mice caused lung inflammation and disruption of Th17/Treg cells development, and altered the composition of gut microbiota through IL-22 induced overexpression of Reg Ⅲγ, leading to intestinal immune injury and disruption of gut microbiota. This research reveals that IL-22 may be the link between the lung and gut. This study may provide a new insight into the intestinal symptoms caused by RSV and other respiratory viruses and the connection between the lung and gut axis, as well as new therapeutic ideas for the treatment of RSV-infected children.

**KEYWORDS**    gut microbiota, IL-22, regenerating islet-derived protein, respiratory syncytial virus

Respiratory syncytial virus (RSV) belongs to the genus of lung viruses that lead to pneumonia (1). It is an enveloped, nonsegmental single-stranded negative-sense RNA virus and is considered the most common pathogen causing severe lower respiratory tract infections in infants worldwide (2, 3). Every year, RSV causes 33 million cases and 3.4 million hospitalizations worldwide in children under 5 years of age (4). In the United States, RSV causes 85,000–144,000 hospitalizations (5) and approximately $2.6 billion in healthcare costs each year (6). Currently, the FDA-approved RSV vaccine Arexvy (RSVPreF3 OA/GSK3844766A) is the first RSV vaccine approved in the United States for the prevention of lower respiratory tract disease caused by RSV infection.

Address correspondence to Shenghai Huang, huangshh68@aliyun.com, Mingwei Chen, chmw1@163.com, or Maozhang He, jnzd_hemaozhang@hotmail.com.

Jiling Liu, Yixuan Huang, and Nian Liu contributed equally to this article. The author order was determined based on their contribution to the article.

The authors declare no conflict of interest.

See the funding table on p. 13.

However, it is only applicable to people over 60 years old (7). In recent years, the role of gut microbiota in airway diseases has received increased interest. The gut microbiome and microbiota-derived metabolites related to RSV infection have been studied in animal models (8, 9). Novel COVID-19 infections are characterized by symptoms such as fever, cough, malaise dyspnea, and gastrointestinal symptoms such as nausea, vomiting, diarrhea, and abdominal pain in some infected individuals. All of the above studies have shown that there is an intestinal microecological imbalance in patients with viral infections, which is manifested by a significant reduction in bacteria with anti-inflammatory activity and beneficial in maintaining human immunity, and by a positive correlation between conditionally competent pathogens and the severity of the disease. This interaction between the gut microbiota and the lungs has been described as the lung-gut axis. The lungs and the large intestine are physiologically and pathologically interconnected and interact with each other, and the emergence of lung disease and intestinal or intestinal disease and lungs, and the diseases of the lungs and intestines can be transformed into each other and even form a vicious circle. In the context of respiratory diseases, most studies have focused on how the gut microbiota influences immune responses in the airways (10), and few studies have investigated the intrinsic mechanisms of the immune response in the respiratory tract affecting intestinal immunity and the intestinal flora. In this paper, we focus on specific components of the infectious factor-immunity-microbiota axis, providing insights into the links between RSV-induced pulmonary immunity, intestinal immunity, and intestinal flora, which could be important for better development of potential preventive or therapeutic tools targeting major infections in children.

More recently, evidence has pointed out the role of two T-cell subsets determining the nature of the immunological response and the severity of RSV infection, namely, interleukin (IL) 17-producing T helper 17 cells (Th17) and regulatory T cells (Treg) (11). Th17 cells mainly secrete proinflammatory cytokines such as IL-17, IL-21, and IL-22 (12, 13). Retinoic acid-related Orphan receptor-T (ROR-γt) directly determines Th17 differentiation (14). Forkhead transcription factor 3 (Foxp3) enhances the immunosuppressive effect of Treg cells (15). The imbalance between Th17 and Treg cells substantially contributes to intestinal immune disturbance and subsequent tissue injury in ulcerative colitis (16). The gut microbiota has been demonstrated to interact with immune cells and to modulate specific signaling pathways involving both innate and adaptive immune processes (17, 18). Th17/Treg cell imbalance and intestinal flora imbalance are common pathogeneses of a variety of diseases (19, 20). Experiments have revealed that influenza virus-infected mice experienced intestinal injury when lung injury occurred, suggesting that influenza virus infection altered the gut microbiota composition (21). Similarly, whether RSV infection induces intestinal lesion and pathogenesis in mice is worth studying. Furthermore, no previous report has described an association between Th17/Treg cell imbalance and intestinal immunity dysregulation in RSV-infected mice.

Regenerating islet-derived (Reg) proteins are small secretory proteins that function as downstream effectors of IL-22. Studies have shown that the production of Reg proteins significantly inhibits inflammatory cell infiltration and promotes tissue repair (22, 23). RegIIIγ, a major member of the family, can be used as an antimicrobial peptide to control the translocation of the intestinal flora and prevent pathogens from infecting the intestine (24–26). IL-22 is an immune cell-produced cytokine that regulates the function of epithelial cells and modulates the epithelial immune microenvironment. In addition, IL-22 induces the secretion of RegⅢγ (27). These findings suggest that IL-22 has a regulatory effect on the expression of the RegIIIγ gene and may play an important role in intestinal immunity and the intestinal flora.

In summary, we hypothesize that RSV infection leads to an imbalance in the development of Th17/Treg cells in the lungs, which affects intestinal immune cell differentiation and alters gut microbiota homeostasis through the excessive release of IL-22 in lung tissue stimulating the expression of RegⅢγ in intestinal tissue via the circulation.

## MATERIALS AND METHODS

### Cells and virus

Human laryngeal epithelial cells (HEp-2) and RSV-Long strains were preserved by the Department of Microbiology, Anhui Medical University. $TCID_{50}$ was determined for viral titers, and $2.5 \times 10^{-6.79}$/0.1 mL was measured by Reed-Muench method in HEp-2 cells.

### Experimental design and sample collection

Seven-day-old suckling mice were randomly divided into three groups of 24 mice each: normal control group, phosphate-buffered saline (PBS) control group, and RSV infection group. For normal control group mice were left untreated, and PBS control group mice were subjected to PBS nasal drip at the same dose as the RSV-infected group mice. For the RSV infection group, the mice were infected with 20 µL of virus suspension by slow nasal drip. The infection cycle lasted 1 week, with RSV infection on day 1 as the benchmark. The mice were weighed daily, then three suckling mice per group randomly selected suckling mice were dissected to collect their lung tissue, small intestine and rectal segments. The left lung was placed in 4% paraformaldehyde for pathological analysis of sections, and the right lung, small intestine, and colon segments were placed in sterile Eppendorf (Ep) tubes for enzyme-linked immunosorbent assay (ELISA) and Real-time quantitative PCR (qRT-PCR) experiments. The rectal segment of suckling mice was rapidly placed into a sterile Ep tube filled with liquid nitrogen. The specimens were sent to Shanghai Peisenol Biotechnology Co., Ltd. for microbial community diversity analysis.

### Real-time quantitative PCR assay (SYBR Green)

Total RNA was extracted from mouse lung tissues using TRIzol reagent. Total RNA was reversely transcribed into cDNA using a commercial reverse-transcription experimental tool kit (TaKaRa, Japan). Gene analysis by qRT-PCR was based on a TaKaRa PCR Kit. GAPDH and β-actin were used as standard internal controls. The sequence of each gene primer is shown in Table 1.

### ELISA assay for IL-10, IL-17, IL-22, IFN-γ, and Reg ⅢY

According to the instructions of this experiment, the levels of IL-10, IL-17, IL-22, IFN-γ, and RegⅢγ were measured. Briefly, the supernatant of lung and intestine tissue at different time points from each group was collected, and the levels of IL-10, IL-17, IL-22, IFN-γ, and Reg Ⅲ γ were determined using high-sensitivity quantitative ELISA kits (Shenzhen Xinbosheng Biological Technology Company).

### Lung histopathological examination

Mouse lungs were preserved in 4% paraformaldehyde overnight, dehydrated through a graded alcohol series, embedded in paraffin, cut into 5-µm-thick sections, and stained with hematoxylin and eosin (H&E). The slides were examined by light microscopy.

### Microbial community analysis

#### *Extraction of total DNA from rectal content, establishment of PCR amplification, and sequencing database*

The rectal segment of suckling mice was rapidly placed into a sterile Ep tube filled with liquid nitrogen. The specimens were sent to Shanghai Paisenol Biotechnology Co., Ltd. for testing and analysis. DNA was extracted from each group of specimens using a DNeasy Power Water Kit DNA Extraction Kit (MoBio, USA). To characterize the taxonomic profile of the gut microbial community, the V3-V4 region of the 16S rRNA gene was amplified from the extracted total DNA. After PCR amplification, the purified

**TABLE 1** Primer sequences for qRT-PCR assay

|         | Gene    | Primer (5′–3′)                   |
|---------|---------|----------------------------------|
| ROR-γt  | Forward | 5′-GCCTCCTGCCACCTTGAGTAT-3′      |
|         | Reverse | 5′-CAAGAGTAAGTTGGCCGTCAG-3′      |
| Foxp3   | Forward | 5′-TGGTTTACTCGCATGTTCGC-3′       |
|         | Reverse | 5′-ACTGCTCCCTTCTCGCTCT-3′        |
| GAPDH   | Forward | 5′-AGGCCGGTGCTGAGTATGTC-3′       |
|         | Reverse | 5′-GGCGGAGATGATGACCCTT-3′        |
| RSV-F   | Forward | 5′-AAACTGCACACATCCCCTCTAT-3′     |
|         | Reverse | 5′-CAGTACCATCCTCTGTCGGTT-3′      |
| β-actin | Forward | 5′-CCAGAGCAAGCGAGGTATCC-3′       |
|         | Reverse | 5′-GCCACACGCAGCTCATTGTA-3′       |

DNA samples were digested with a restriction enzyme and quantified based on the enzyme marker. The specific primers used are listed in Table 2.

An Illumina TruSeq DNA Kit was used to establish a sequencing genome library, followed by microbial diversity analysis. DNA fragments were fragmented by high-pressure nitrogen or Covaris (frozen cell fragmentation) technology, and DNA fragments with prominent ends were modified by the 3′–5′ exonuclease and polymerase in End Repair Mix. After modification, base A was introduced to ensure that base T of the joint was complementary to prevent the connection between fragments. The labeled adaptor was connected with DNA fragments by ligase, and suitable DNA fragments were selected for gel purification. DNA fragments with labeled junctions were amplified by PCR and picogreen and a fluorescence spectrophotometer was used to quantify the library. The quality of PCR-enriched fragments was controlled based on the results of analysis with an Agilent 2100 to verify the size and distribution of DNA library fragments. The homogenized multisample DNA library was diluted to 10 nM and mixed in equal volumes. Finally, the homogenized library was diluted to 4–5 pM and sequenced.

## 16S amplicon sequencing data analysis

After DNA extraction and sequencing, the raw paired-end reads underwent a data curation pipeline, which included the removal of low-quality reads (Qiime2 2020.8). Subsequently, the remaining sequences were assigned to their respective samples based on barcode matches, and the barcode and primer sequences were subsequently trimmed. The sequences underwent denoization using the DADA2 method, and reads were classified using the Silva reference database (version 138). A total of 1,324,704 sequence reads were analyzed, with an average of 36,316 reads per sample (range: 23,256–45,176). Alpha and beta diversity were computed using Qiime2 2020.8. Principal coordinate analyses (PCAs) based on the Bray-Curtis distance were conducted to assess beta diversity. Chao1 and Shannon indexes were calculated to characterize alpha diversity. Differential analysis was performed utilizing the Wilcoxon rank sum test. Spearman's correlations were used to analyze the relationships between bacterial taxa and metabolites. Correlations were considered significant when the adjusted $P$ values were less than 0.05 after correction for the false discovery rate, using the Benjamini-Hochberg procedure. According to the results of 16S rRNA, the metabolic function of samples was predicted by PICRUST2, the differentially abundant pathways and the composition of specific pathways were also obtained.

**TABLE 2** The primer sequence and length of the extended product of 16S rRNA

| Gene      | Designated oligonucleotides (5′–3′) | Amplicon length (bp) |
|-----------|-------------------------------------|----------------------|
| 16S V3-V4 | F:5′-ACTCCTACGGGAGGCAGCA-3′         | 480                  |
|           | R: 5′-GGACTACHVGGGTWTCTAAT-3′       |                      |

## Statistical analysis

All data are expressed as the mean ± SEM values. One-way analysis of variance was used to identify statistically significant differences among groups (SPSS 19.0; SPSS, Chicago, IL, USA). $P < 0.05$ was considered to indicate significance.

## RESULTS

### RSV infection causes lung injury in suckling mice

Seven-day-old BALB/c suckling mice were intranasally inoculated with virus, and the flow chart of the experiment is shown in Fig. 1A. Within 1 week after RSV infection, weigh the suckling mice at the same time every day. The body weight of suckling mice was significantly lower after RSV infection than that for the normal control group and PBS control group, with statistical significance from the second day after infection ($P < 0.05$, Fig. 1B). The result suggests that compared with the normal group mice, the growth of infected suckling mice was slower. Moreover, the mental state was relatively depressed. The effect of lung damage caused by viral infection could be observed by the naked eye and verified by H&E staining of lung tissue. Normal group lungs (Fig. 1C, left) were a uniform powdery white with a smooth surface and no swelling or hemorrhagic lesions were illustrated. In the RSV-infected group (Fig. 1C, right), the lungs of suckling mice were dark red, with poor elasticity and patchy hyperemia. Subsequently, pulmonary pathology was dete cted in the normal group mice (Fig. 1D, a) and the mice infected with virus for 1–7 days (Fig. 1D, b–h). The results showed that the alveolar structures of the normal group were clear, the alveolar septa were uniform, and there was no thickening, hyperemia, or infiltration of inflammatory cells. However, in the infected group, the alveolar structure was disordered, the alveolar septa were thickened, and inflammatory cell infiltration was observed in the alveolar cavity and pulmonary interstitial. Pulmonary

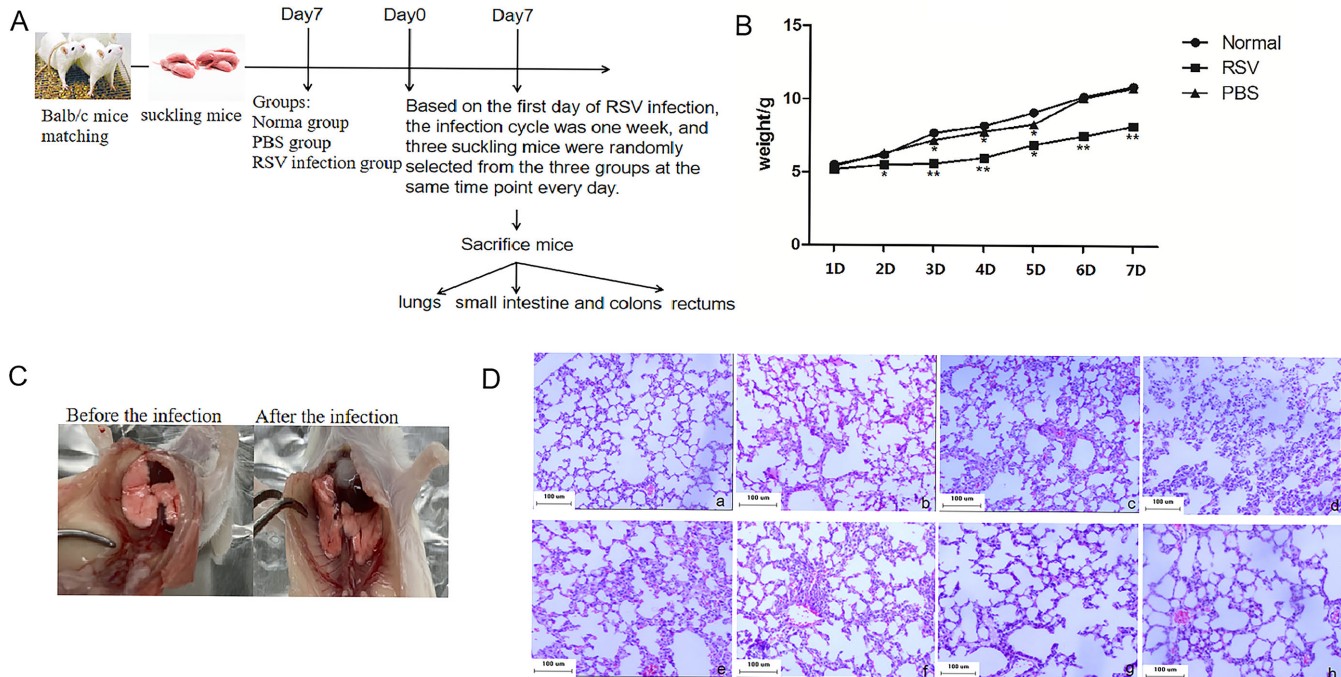

FIG 1 RSV infection can cause lung injury in suckling mice. (A) Seven-day-old BALB/c suckling mice were intranasally inoculated with virus, and a flow chart of the experiment is shown. (B) Monitoring of mouse weight from days 1 to 7. Each data point represents the mean weight (±SD) of all mice from the indicated group. * <0.05, ** <0.01, compared with the normal control group. (C) Macroscopic observation of lung tissue in suckling mice. Left side: Lung tissue of the normal group; Right side: lung tissue of the RSV-infected group. (D) Results of pathological section analysis of lungs from suckling mice (H&E ×100). (a) Pathological sections of lungs in normal group, (b–h) Pathological sections of lungs 1–7 days after RSV infection.

inflammation peaked on the third day (3 days) after infection and then began to resolve gradually. In summary, we have successfully established an RSV infection model in neonatal mice.

## Measurement of ROR-γt and Foxp3 mRNA expression in lung tissue of the control group by qRT-PCR verifies successful negative control modeling

ROR-γt is the most critical regulator of Th17 differentiation (14). Treg cells are characterized by elevated Foxp3 expression (15). As shown in Fig. 2, the mRNA levels of ROR-γt and Foxp3 in the lung tissues of the PBS control group were not significantly different from those of the normal group. Therefore, the negative control group was successfully established, and the subsequent experimental analysis was mainly conducted between the normal control group and the infected group.

## ROR-γt and Foxp3 mRNA expression in the lung and intestine tissues was temporally upregulated after RSV infection

RSV-F gene expression in the lung and intestine tissue of mice was measured by qRT-PCR at 1–7 days (Fig. 3). As shown in Fig. 3A, the F protein mRNA expression of RSV in the lungs and the intestine was not detected in the normal control group, but on the third day after the virus infection, the greatest amount of expression was observed in the lungs of the RSV infection group ($P < 0.01$), onefold increase compared to the first day after infection, and then began to decline. However, no virus was detected in the intestinal tissue of the infected group, indicating that RSV did not directly infect the intestinal tissue. As shown in Fig. 3B and C, the mRNA levels of ROR-γt and Foxp3 in the lung and intestinal tissues were increased in the first 3 days and then decreased, while the mRNA levels of ROR-γt in the lung tissues were highest on the third day after infection, and the mRNA levels of Foxp3 were the highest on the fourth day after infection, approximately 2-fold and 1.5-fold increase, respectively, compared to normal controls. The mRNA levels of ROR-γt in intestinal tissues were the highest on the second day after infection, and the mRNA levels of Foxp3 were the highest on the third day after infection, with statistically significant differences ($P < 0.01$). These results revealed that the expression of Th17 and Treg cells in pulmonary and intestinal tissues showed temporal differences. Moreover, the difference in intestinal tissues is not directly caused by RSV infection.

## The excessive release of IL-22 in lung tissue may stimulate the expression of Reg ⅢγΥ in intestinal tissue via the circulation

Compared with the normal group, the expression of IFN-γ in the lungs and intestines of the RSV-infected group showed a trend of first increasing and then decreasing, reaching

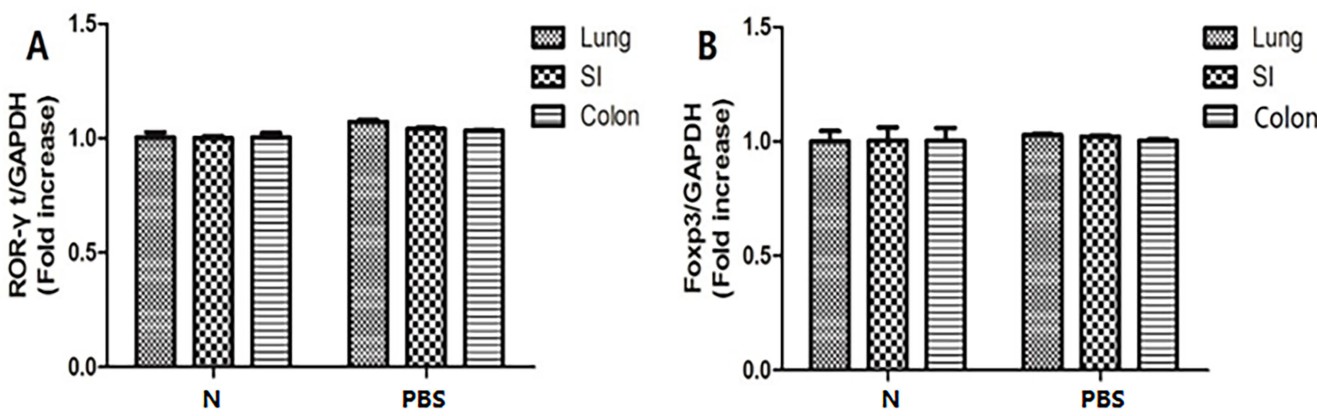

**FIG 2** The mRNA levels of ROR-γt and Foxp3 in the lung and intestine of the normal control group and PBS control group were detected by qRT-PCR. (A) The expression of ROR-γt mRNA. (B) The expression of Foxp3 mRNA.

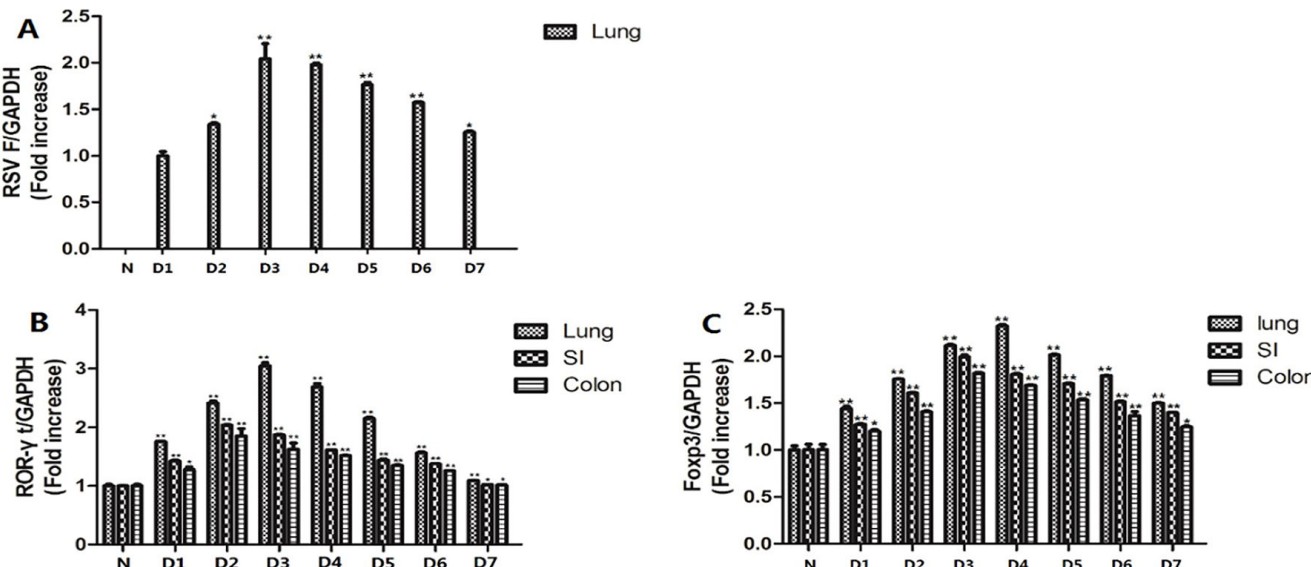

**FIG 3** The expression of RSV-F, ROR-γt, and Foxp3 mRNA in lung and intestinal tissues of normal control and RSV-infected groups at different time points was detected by qRT-PCR. (A) The expression of RSV-F mRNA. (B) The expression of ROR-γt mRNA. (C) The expression of Foxp3 mRNA. * <0.05, ** <0.01, compared with the normal control group.

a maximum value one the third day after infection, with a significant difference ($P < 0.01$, Fig. 4A). The expression level of IFN-γ in the lungs was always higher than that in the intestine, which indicates that the body's immune response is the most active on the third day after infection. Compared with the normal group, the expression of IL-17 and IL-22 increased first and then decreased after RSV infection, peaking in the lung on the third day after infection and in the small intestine and colon on the second day after infection ($P < 0.01$, Fig. 4B and C). The content of IL-10 in the lung was the highest on the fourth day after infection, and the content of IL-10 in intestinal tissues was the highest on the third day after infection, with significant differences ($P < 0.01$, Fig. 4D). The expression of Reg Ⅲγ in the intestinal tissues reached its maximum value ($P < 0.01$, Fig. 4E) on the third day after RSV infection and then began to decline. The results suggested that the excessive release of IL-22 in lung tissues may stimulate the expression of Reg Ⅲγ in intestinal tissues via the circulation, thus affecting the secretion of inflammatory factors by intestinal Th17 and Treg cells.

## RSV infection changed the species composition of the intestinal flora of suckling mice but the not richness and diversity

To elucidate the changes in the intestinal flora after RSV infection, we compared dynamic changes in four genera representative, including their relative abundances and species composition. As shown in Fig. 5A, the alpha diversity analysis showed no significant difference between the normal control group and the RSV infection group, indicating that RSV infection did not cause significant changes in the intestinal flora richness and diversity of suckling mice. As shown in Fig. 5B, the PCA results elucidated that the set of bacteria of the control group and the infection group were completely separated after eliminating outliers based on beta diversity analysis ($P = 0.003$). Jaccard distance matrix variance analysis (Fig. 5C) showed a significant difference between the two groups ($P = 0.0000755$), indicating that the species composition of the rat intestinal flora changed after RSV infection.

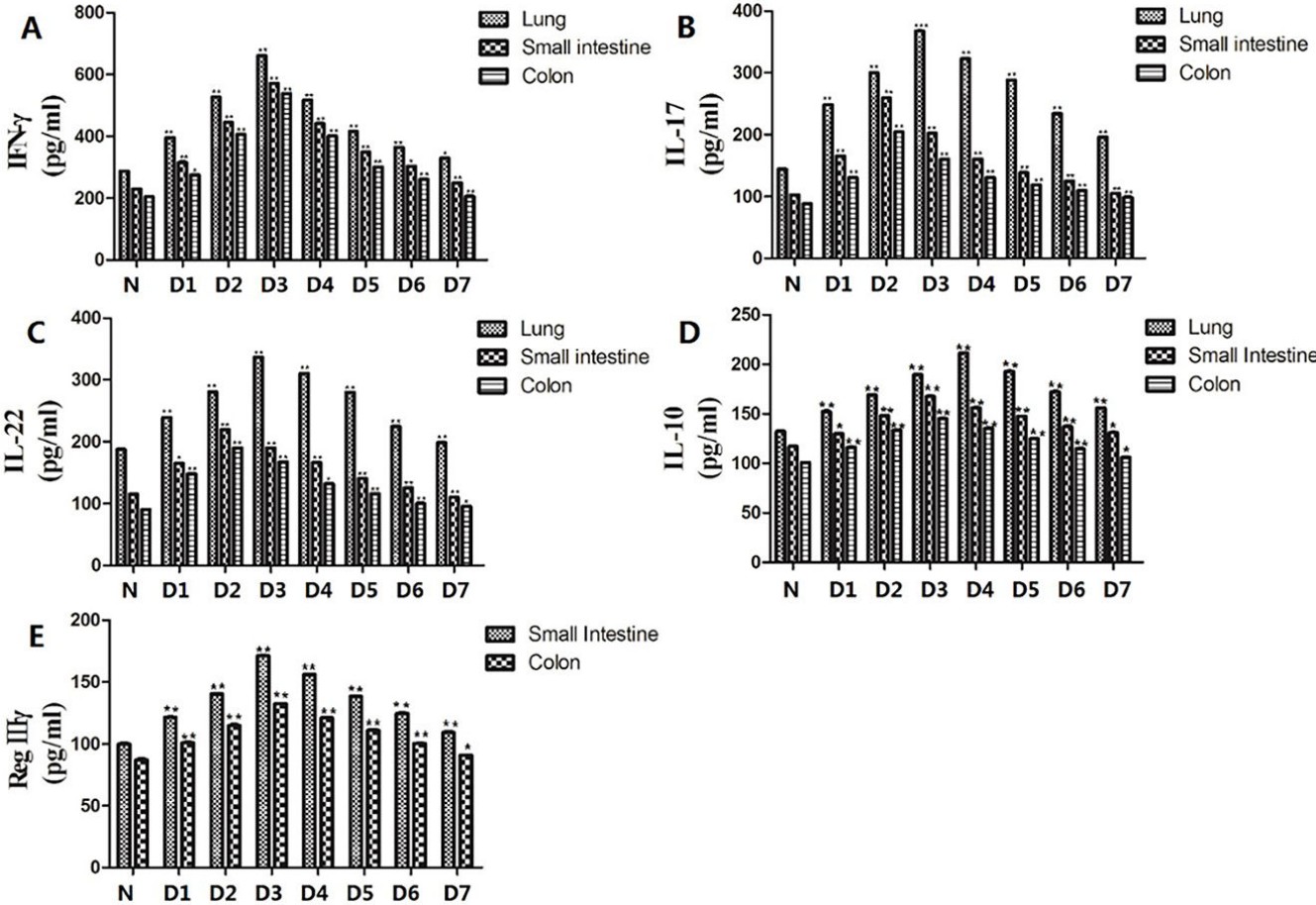

**FIG 4** The levels of IFN-γ, IL-17, IL-22, IL-10, and Reg Ⅲ γ in the supernatant of lung and intestine tissues at different time points were measured by ELISA assay. * <0.05, ** <0.01, compared with the normal control group. All tests were repeated three times. (A) The expression of IFN-γ. (B) The expression of IL-17. (C) The expression of IL-22. (D) The expression of IL-10. (E) The expression of Reg Ⅲ γ.

## Analysis of RSV infection-induced differences in bacterial genera composition

Bacterial genus differences between the normal control group and the RSV-infected group were analyzed. Wilcoxon group difference analysis (Fig. 6A) showed that the bacterial genus enriched in the RSV group was *Aggregatibacter*, while *Proteus* was reduced. Fifteen important bacterial genera were selected for random forest validation (Fig. 6B) and rank among the top of are *Proteus* and *Aggregatibacter*. This finding suggested that RSV infection changes the genus composition.

## Correlation analysis of different bacterial genera and different KEGG pathways

Next, genera and KEGG pathways derived from variance analysis were compiled in tables, and Spearman correlation analysis was applied to derive the relation of KEGG pathways with genera. The results are shown in Fig. 7. The genera of bacteria associated with diversity in metabolic pathways mainly included *Acinetobacter*, *Aggregatibacter*, *Proteus*, *Corynebacterium*, and *Lactobacillus*. The differentially abundant metabolic pathways are mainly reflected by changes in the contents of insulin metabolites, fatty acid metabolites, glycerophosphate metabolites, phosphocreatine metabolites, tryptophan metabolites, and arginine and proline metabolites. The level of tryptophan metabolites showed the most significant difference.

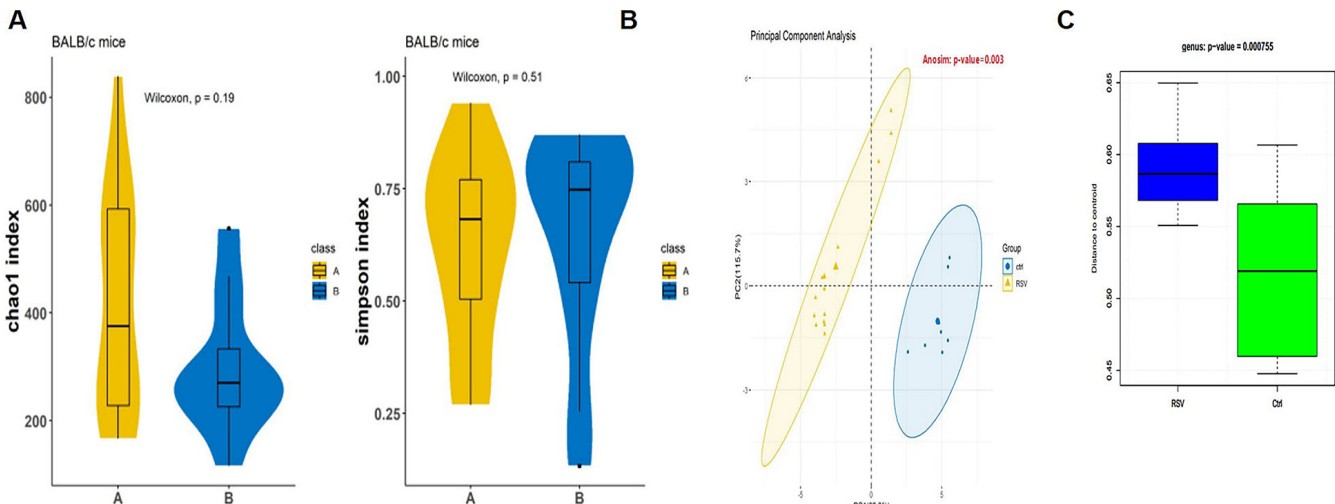

**FIG 5** RSV infection did not cause significant changes in the abundance and diversity of the intestinal flora of suckling mice, but the species composition of the gut bacteria changed. (A) Differential analysis was performed using the Wilcoxon rank-sum test method by calculating the *P* value. (B) PCA. Each point in the figure represents a sample, and the points with the same color came from the same group. The closer the distance between two points is, the smaller the difference in community composition between the two points. When *P* value was <0.05, the difference was significant. (C) Jaccard analysis. The Jaccard distance matrix variance analysis showed a significant difference between the two groups after RSV infection, which correspond to changes in the species composition of the mouse intestinal flora.

## DISCUSSION

RSV infection in children has become a global concern due to its high incidence, high mortality, and specific clinical treatment. It causes extensive clinical manifestations including upper and lower respiratory tract infections (28). A study reported that the human Boca virus is usually detected in children with acute respiratory or gastrointestinal symptoms and usually coinfects with RSV (29), which suggests that RSV infection alone may cause intestinal symptoms in addition to respiratory reactions. However, the study did not address the effect of RSV on the intestine alone. Recent studies have also shown that COVID-19 patients experience gastrointestinal symptoms such as diarrhea and vomiting along with the most common respiratory symptoms, suggesting gut involvement during infection (30–32). Several studies have demonstrated the relevance of the intestinal microbiota in inflammatory and infectious conditions, including asthma, colitis, and bacterial and viral infections (9). The above studies suggested a certain relationship between RSV infection and intestinal homeostasis. The idea of whether RSV infection causes intestinal immune damage and intestinal flora disorder in addition to lung tissue damage prompted us to further study the relationship between the lung and intestinal axis.

It was found that the Treg cell-induced immune response ensures effective virus clearance and reduces clinical symptoms by inhibiting the Th2 cell immune response, while the Th17 cell-induced immune response is believed to prevent effective virus clearance and further enhance the Th2 cell immune response, leading to severe clinical manifestations (33). The research found that infection of human bronchial epithelial cells with RSV stimulates T cells to undergo differentiation into Th17 cells and inhibits their differentiation into Treg cells (34). In the current study, to better match the age of RSV infection in humans and to learn how Th17 and Treg cells change after RSV infection, 7-day-old lactating mice with undeveloped immune systems were employed. ROR-γt is an essential transcription factor in Th17 cell development, and Foxp3 is a critical transcription factor in the development of Treg cells (14, 15). Due to the small size of the suckling mice, the peripheral blood was not collected for flow cytometry experiments, so the changes in the ratio of the two kinds of immune cells could not be directly assessed, but their level based on the ROR-γt and Foxp3 mRNA levels could be determined. In this

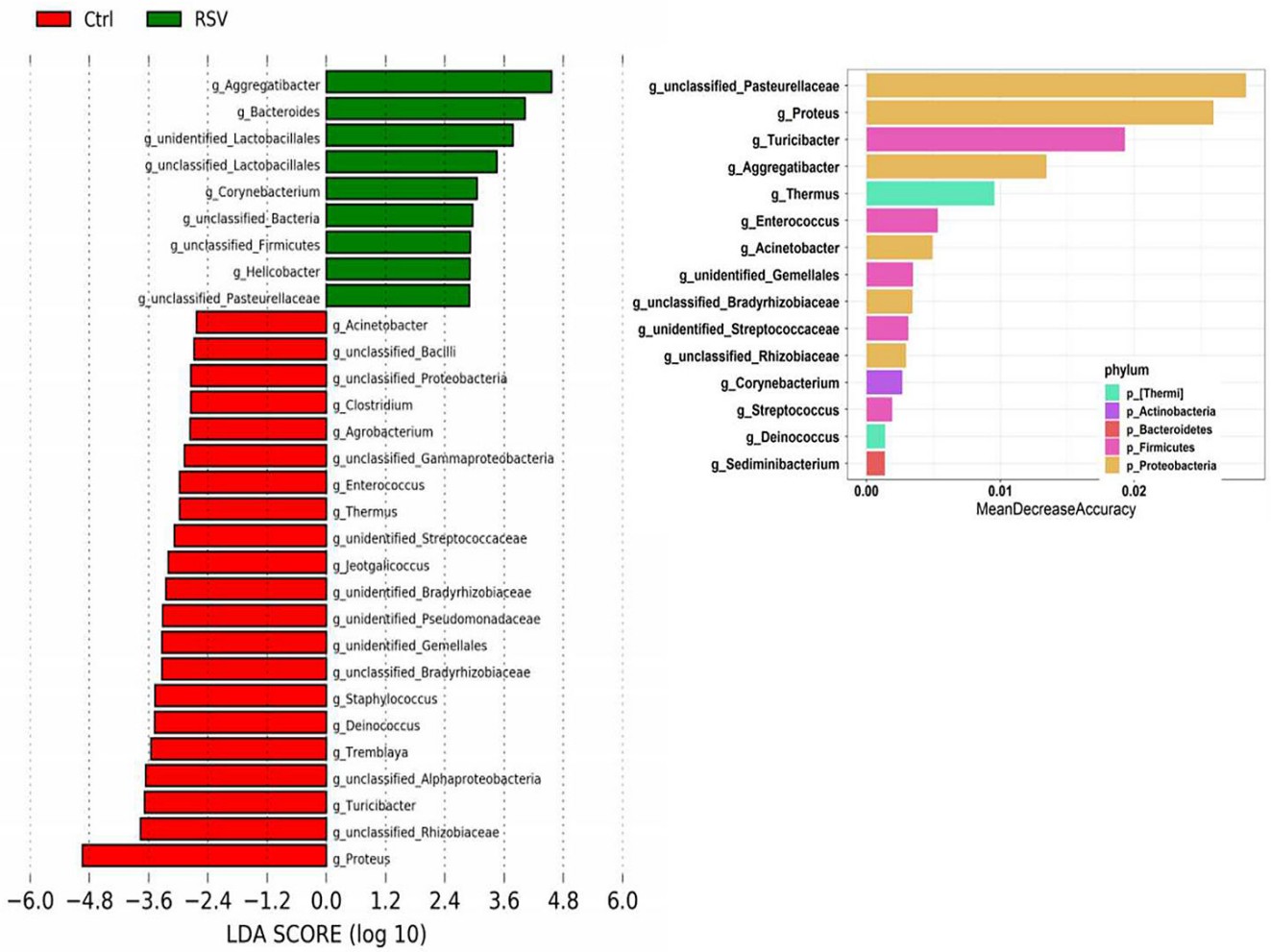

**FIG 6** Analysis of the difference in bacterial abundance. (A) LDA value distribution histogram showing the results of Wilcoxon group difference analysis between the control group and the RSV infection cohorts. The color of the histogram represents the group, and the length of the histogram represents the LDA score, that is, the degree of influence of distinct species between different groups. (B) Random forest verification. Screening of differentially abundant microbiota by random forest analysis. Rank among the top of are *Proteus* and *Aggregatibacter*.

study, RSV-infected neonatal mice showed lung inflammation and changes in lung Th17 and Treg cell levels. Moreover, Th17 cells in the lung tissues after RSV infection reached the maximum level on the third day, while Treg cells reached the maximum level on the fourth day, and the expression level of Th17 cells was higher than that of Treg cells. The finding indicates that viral infection affects the development and differentiation of Th17 and Treg cells and affects the Th17/Treg balance. RSV first stimulates the expression and release of inflammatory cytokines in Th17 cells, leading to an inflammatory response in the body. Subsequently, Treg cells begin to express and release anti-inflammatory cytokines, providing protection for the body and clearing the virus. The pathological changes in the lungs also confirmed this conclusion. The above studies showed that RSV infection in BALB/c neonatal mice caused an imbalance in the proportion of Th17/Treg cells, which peaked on the third day after infection.

Reg Ⅲ γ is an antimicrobial peptide that is usually expressed by intestinal epithelial cells (IECs) and plays an important role in intestinal homeostasis and intestinal microflora control (35). Zheng et al. discovered that IL-22 induces the expression of Reg Ⅲ β and RegⅢγ in IECs, inhibits *Citrobacter rodentium* infection in the colon, and protects colon epithelial cells from damage (36). A clinical study showed that the level of IL-22 in

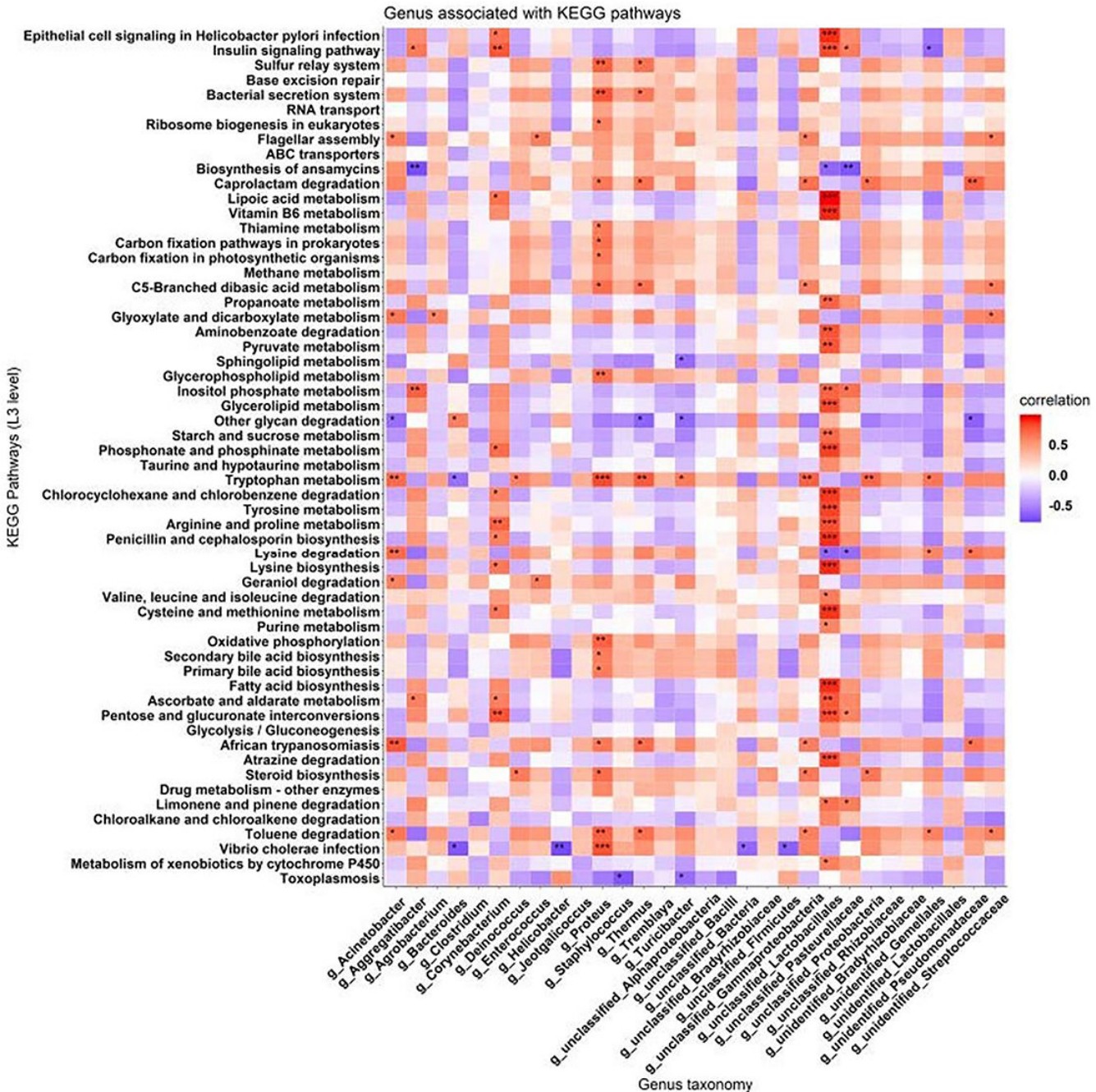

**FIG 7** Correlation analysis of differentially abundant genera and differentially abundant KEGG pathways. KEGG pathway and genus taxonomy analysis show the pathways/genera with consistent changes in abundance [defined here as significant alterations (adjusted $P < 0.05$)]. Pathways/genera that showed increased and decreased abundance are shown as red and blue squares, respectively; square colors indicate the enrichment scores.

the peripheral blood serum of COPD patients was significantly higher than that of the normal control subjects (37). Lin et al. found that the plasma IL-22 levels in patients with acute myocardial infarction (AMI), stable angina (SA), and unstable angina (UA) were significantly higher than those in the normal subjects (38). The finding concluded that IL-22 released into the blood plays a corresponding role when the body is in a state of disease. Brand et al. confirmed that the IL-22 receptor exists on the intestinal surface, and IL-22 binding activates Reg Ⅲ γ expression in intestinal epithelial cells through the STAT3 pathway (39). In summary, IL-22 plays an important role in Reg Ⅲ γ expression.

In addition, in lupus-prone mice, activated B cells inhibit Th17 cell differentiation and immune function via IL-22 (40). In our experiment, the level of Reg Ⅲ γ in intestinal tissue reached a maximum on the third day after infection, while the level of IL-22 reached a maximum on the second day after infection. Although IL-22 upregulated the expression of Reg Ⅲ γ, the maximum levels of the two were different at different time points. Based on the above literature, we speculate that IL-22 released from lung tissue accumulates in the circulation to stimulate the expression of Reg Ⅲ γ in the intestinal mucosa, but it also inhibits the release of IL-22 from Th17 cells in the intestinal tissue itself. Based on the results of qRT-PCR, we also found that the expression of Th17 cells in lung tissues reached a maximum value on the third day after viral infection. However, in intestinal tissues, the maximum level was reached on the second day after viral infection. The above findings suggested that the imbalance in the Th17/Treg cell ratio in the lung induced by RSV infection in BALB/c neonatal mice results in the release of excessive IL-22, which mediates intestinal immune injury through circulating IL-22. To confirm this hypothesis, injection of IL-22 inhibitors into the animals and examination of changes in the two immune cell types in the intestine will be conducted in the future.

The development of the intestinal immune system and intestinal microbial flora complement each other to maintain the balance of the intestinal environment and host health. Qin et al. analyzed the number of flora genes in the feces of patients with inflammatory bowel disease (IBD) through high-throughput metagenomics and found that it was 25% lower than that of healthy people (41). Garrett et al. found that the load of *Proteus* and *Streptococcus pneumoniae* in the feces of spontaneous ulcerative colitis (UC) mice was significantly increased, and after the intestinal flora of UC mice was transplanted into normal mice, the latter also suffered from UC (42). Among the sections of the intestinal tract, the colon and rectum contain the most bacteria. IL-22 is expressed in a variety of tissues, including lung and gastrointestinal tract. IL-22 receptors are expressed on the stromal and epithelial cells of these tissues. Since IL-22 is produced at the site of inflammation, it may mediate physiological responses to repair local tissue damage, or leads to pathophysiological inflammation (43). In this experiment, although RSV infection did not significantly change the abundance and diversity of the flora in the intestinal tract of neonatal mice, it significantly changed the composition of the flora. Research showed that during *S. Typhimurium* infection, IL-22 is instead exploited by the pathogen to colonize the gut to high levels (44). In our study, IL-22 binds to receptors on intestinal mucosal epithelial cells with blood circulation and stimulates the expression of Reg Ⅲ γ in the intestinal mucosa. Reg Ⅲ γ further changes the species composition of intestinal flora in suckling mice, causing changes in the ratio of *Aggregatibacter* and *Proteus* and the imbalance of Th17/Treg cell ratio in the intestine. Upon further investigation of the effect of changes in the flora on neonatal mice by studying the flora metabolism pathways, we found that tryptophan metabolism may affect the body's immune system. Studies have shown that tryptophan has anti-inflammatory effects in mammals, and mice fed with a low-tryptophan diet are more likely to produce inflammatory responses. In addition, tryptophan metabolites regulate the production of IL-22 through aryl hydrocarbon receptor (AhR), which plays a key role in immune homeostasis. Subsequently, we further optimized the flora analysis and conducted experiments with tryptophan supplementation *in vitro* to explore its possible role in the body.

The innovation of this experiment is that we explored the connection of the lung-gut axis mechanism and discovered that IL-22 may bridge the lung and intestine. Excessive release of IL-22 from the lungs binds to the IL-22 receptor on the intestinal surface through blood circulation and induces the overexpression of Reg Ⅲ γ, and the species composition of intestinal flora in suckling mice was further changed, which will also inhibit the release of IL-22 from Th17 in the intestinal tissues itself. However, there are several limitations in this study. For example, the fecal samples of suckling mice are difficult to collect, so we collected the rectal tissue of suckling mice, which contained the immature feces.

This research takes the lung inflammation caused by RSV infection in BALB/c sucking mice as an entry point to explore the relationship between the effects of RSV on lung immunity, intestinal immunity, and intestinal flora, which is expected to provide new ideas and strategies for the prevention and treatment of respiratory tract infections. In addition, the study expands the understanding of the mechanism underlying intestinal damage caused by respiratory virus infection, and is of guidance for research into the mechanism of intestinal damage caused by other respiratory viruses such as influenza virus and SARS-CoV-2.

## ACKNOWLEDGMENTS

This study was supported by the Natural National Science Foundation of China (Nos. 81974306 and 82302568), Major Project of Natural Science Research of Anhui Education Department (No. KJ2019ZD23), Research Fund of Anhui Institute of Translational Medicine (Nos. 2017zhyx26 and 2021zhyx-B07), National First-Class Undergraduate Program Construction Point (Biotechnology), and Projects of Education Quality Engineering of Anhui Province (No. 2021jcxkpy013).

## AUTHOR AFFILIATIONS

[1]Department of Microbiology, The Key Laboratory of Microbiology and Parasitology of Anhui Province, The Key Laboratory of Zoonoses of High Institutions in Anhui, School of Basic Medical Sciences, Anhui Medical University, Hefei, Anhui, China
[2]College of Life Science, Hebei University, Baoding, Hebei, China
[3]Department of Endocrinology, The First Affiliated Hospital of Anhui Medical University, Hefei, Anhui, China
[4]Department of Critical Care Medicine, The First Affiliated Hospital of Anhui Medical University, Hefei, Anhui, China
[5]School of Nursing, Anhui Medical University, Hefei, Anhui, China
[6]School of Life Sciences, Anhui Medical University, Hefei, Anhui, China

## AUTHOR ORCIDs

Jiling Liu http://orcid.org/0009-0003-3110-4451
Maozhang He http://orcid.org/0000-0002-0743-7323
Mingwei Chen http://orcid.org/0000-0002-8439-0469
Shenghai Huang http://orcid.org/0000-0002-5699-8928

## FUNDING

| Funder | Grant(s) | Author(s) |
|---|---|---|
| MOST \| National Natural Science Foundation of China (NSFC) | 81974306 | Shenghai Huang |
| the Natural National Science Foundation of China | 81974306, 82302568 | Maozhang He |
| | | Shenghai Huang |
| Major Project of Natural Science Research of Anhui Education Department | KJ2019ZD23 | Shenghai Huang |
| Research Fund of Anhui Institute of Translational Medicine | 2017zhyx26, 2021zhyx-B07 | Shenghai Huang |
| National First-Class Undergraduate Program Construction Point (Biotechnology) | 2021jcxkpy013 | Shenghai Huang |
| Projects of Education Quality Engineering of Anhui Province | 2021jcxkpy013 | Shenghai Huang |

## AUTHOR CONTRIBUTIONS

Jiling Liu, Data curation, Writing – original draft, Methodology | Yixuan Huang, Data curation, Methodology | Nian Liu, Methodology | Huan Qiu, Formal analysis, Supervision | Xiaojie Liu, Methodology | Maozhang He, Formal analysis, Supervision | Mingwei Chen, Conceptualization | Shenghai Huang, Conceptualization, Writing – review and editing.

## ETHICS APPROVAL

BALB/c mice were purchased from Zhejiang Viton Lihua Laboratory Animal Experimental Company under animal license number SCXK (Zhejiang) 2019-0001. The animals were kept in the Anhui Medical University Testing Center P2 laboratory. The laboratory animal license number was SCXK (Zhejiang) 2019-0034, SPF grade.

## ADDITIONAL FILES

The following material is available online.

### Open Peer Review

**PEER REVIEW HISTORY (review-history.pdf).** An accounting of the reviewer comments and feedback.

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
