## [Reviewer comments · Microbiology Spectrum]

Microbiology Spectrum

The imbalance of pulmonary Th17/Treg cells in BALB/c suckling mice infected with respiratory syncytial virus mediated intestinal immune damage and gut microbiota changes

Jiling Liu, Yixuan Huang, Nian Liu, Huan Qiu, Xiaoyan Zhang, Xiaojie Liu, Maozhang He, Mingwei Chen, and Shenghai Huang

Corresponding Author(s): Shenghai Huang, Anhui Medical University

Review Timeline:

Submission Date:	October 10, 2023
Editorial Decision:	January 16, 2024
Revision Received:	March 16, 2024
Accepted:	April 2, 2024

Editor: Dhammika Navarathna

Reviewer(s): The reviewers have opted to remain anonymous.

Transaction Report:

DOI: <https://doi.org/10.1128/spectrum.03283-23>

Re: Spectrum03283-23 (The imbalance of pulmonary Th17/Treg cells in BALB/c suckling mice infected with respiratory syncytial virus mediated intestinal immune damage and gut microbiota changes)

Dear Dr. Shenghai Huang:

Thank you for the privilege of reviewing your work. Below you will find my comments, instructions from the Spectrum editorial office, and the reviewer comments.

Revision Guidelines

Sincerely,
Dhammika Navarathna
Editor
Microbiology Spectrum

Reviewer #1 (Comments for the Author):

In this manuscript, it describes that RSV infection of seven-day-old suckling mice causes an imbalance of lung and intestinal Th17/Treg cell immunity and disturbances in gut flora, and that IL-22 may be responsible for this phenomenon. The innovation of this experiment is that we explored the connection of the lung-gut axis mechanism and discovered that IL-22 may bridge the lung and intestine. However, there are several major and minor drawbacks in the presented work.

Major questions:

Comment #1

Why aren't the diagrams in Figures 2 and 3 together?

Comment #2

In the manuscript, the authors propose primer sequences including 16S total length, 16S V3 V4, 16S V4.16S V4 V5, have you all tested them?

Comment #3

The authors well presented that RSV infection in BALB/c mice caused lung immune imbalance, intestinal immune imbalance and intestinal flora disorders. By establishing a model of RSV-infected BALB/c mice, and doing qPCR, HE and intestinal flora sequencing experiments, the authors revealed that RSV infection not only caused an imbalance of Th17/Treg cells in the lungs of the mice, but also caused an excess of IL-22 to be released from the lungs through the circulation. Excessive IL-22, which binds to the IL-22 receptor on the intestinal surface and induces intestinal Reg III γ overexpression, also leads to the release of excessive IL-22 from the lungs through blood circulation, and the binding of IL-22 to the IL-22 receptor on the intestinal surface, which induces impaired Reg III γ overexpression, impaired development of intestinal Th17/Treg, and altered composition of the gut microbiota. The language may need to be further modified and the logic checked.

Comment #4

In the "Experimental design and sample collection" section of the article, the experimental grouping process should be described in detail, it is not very clear how the groups are grouped and how each group is handled.

Comment #5

Figure 5 is not clearly annotated. Please revise it.

Comment #6

It makes sense for the authors to write this article, but there is a lack of mechanistic validation, so there should be provide elucidation about the existing limitations of this work and how to follow it up.

Minor questions:

Comment #1

The fonts of some pictures in the article are not very beautiful, such as Figure 1C. Please adjust the layout of the diagram.

Comment #2

There are a few punctuation omissions in the article, for example, the page 1 correspondence section, so please double-check the calibration.

Comment #3

Where the text states that there has been previous research, please add the appropriate literature cited.

Comment #4

Wouldn't this phrase "the first several of which" be better expressed in a different way?

Comment #5

**p < 0.01 preceded by *, proposed to be deleted

Comment #6

Acknowledgement section is a little inappropriate. This part should make a revision.

Comment #7

The keywords should be separated by commas. Please revise it.

Reviewer #2 (Comments for the Author):

Figure 1. While the authors claim success in establishing an RSV infection model in neonatal mice, it is essential to acknowledge Harker et al.'s previous findings on exacerbated disease upon reinfection also in neonatal mice (<https://doi.org/10.1128/jvi.02620-13>).

Figure 2.

The claim of successfully establishing a negative control group is unclear; the subsequent experimental analysis predominantly

compared the normal control group with the infected group. However, for a more meaningful comparison, it would be more logical to assess differences between the RSV-infected group and the PBS control group. Clarifying this aspect would enhance the clarity and validity of the experimental design and analysis.

Figure 3. When detailing the kinetics of ROR- γ t and Foxp3 expression in lung and intestinal tissues, it is advisable to specify the fold increase compared to the control rather than relying on statistical significance (P value) for clarity in interpretation. This approach provides a more direct and understandable representation of the changes in gene expression over time.

Figure 4.

IL-22 serves dual roles, acting as both a protective and proinflammatory factor, often indicating a mechanism for lung repair. Its function hinges on downstream expression, influenced by chemokines and inflammatory cytokines like IL-1 β , IL-17, and TNF- α . To establish meaningful connections with RegIII γ in intestinal tissue, it is crucial to profile major cytokines originating from the lungs. Examining their profiles in the bloodstream could provide valuable insights, offering hints about potential relationships or correlations. Therefore more experiments on RT-PCR, ELISA targeting the major cytokines is desired.

Figure 5. Authors mentions about "indicating that the species composition of the rat intestinal flora changed after RSV infection". Did the author use rats for experimentation as well?

The impact of inflammatory cytokines on gut microbiota is established, and daily diet significantly shapes the animal gut microbiome. In disease conditions, such as the inflammation observed in suckling mice, altered dietary patterns may contribute to changes in the gut microbiome. Considering these factors, the observed alterations in the bacteriome profile may not solely be attributed to the RSV-induced cytokine response. To enhance understanding, it is recommended that the authors incorporate another respiratory pathogen model for cross-referencing changes in the bacteriome profile.

81 Meishan Road, Hefei, Anhui 230032, China

March 16, 2024

Editorial Staff, *Microbiology Spectrum*

Dear editor and reviewers,

In response to your decision on our submission of "The imbalance of pulmonary Th17/Treg cells in BALB/c suckling mice infected with respiratory syncytial virus mediated intestinal immune damage and gut microbiota changes" to *Microbiology Spectrum* (Manuscript ID Spectrum03283-23).

Thank you for considering our manuscript and providing insightful comments and recommendations, which have not only made this work scientifically more accurate and sound, but also helped to improve its readability to a broad audience. Here we are submitting the revision of the manuscript, along with this responding letter addressing the suggestions.

We have updated the manuscript, especially for the issues raised by reviewers, and proof-read again to make the descriptions to be more precise. Again, thank you and the anonymous reviewers for the tremendous efforts helping us improve this manuscript.

We have to mention that in order to address reviewers' comments, we have added more discussion and explanation in main text. We have also revised the relevant sentences in the manuscript to ensure that there are no errors, however, due to space constraints, we are unable to go into depth on each of the questions raised (although we have answered all of them).

We guarantee any part of the paper has not been published or is under consideration for publication elsewhere, including on the Internet.

We thank you for your time and consideration, and look forward to future correspondence.

We hope you will find our updated manuscript to be suitable for *Microbiology Spectrum*. Thank you again for the valuable and constructive suggestion!

Yours Sincerely

Best Regards

Shenghai Huang, Mingwei Chen, Maozhang He

Responses to comments:

Reviewer 1:

Comments to the Author

Overall Comment

In this manuscript, it describes that RSV infection of seven-day-old suckling mice causes an imbalance of lung and intestinal Th17/Treg cell immunity and disturbances in gut flora, and that IL-22 may be responsible for this phenomenon. The innovation of this experiment is that they explored the connection of the lung-gut axis mechanism and discovered that IL-22 may bridge the lung and intestine. However, there are several major and minor drawbacks in the presented work.

Answer: We thank reviewer for these valuable and constructive comments. According to the suggestions raised by reviewer, we have provided more explanation and modified our manuscript. Please check the latest manuscript. We have given the detailed answers for each comment as follows, please check. Thanks again.

Major questions:

Comment #1

Why aren't the diagrams in Figures 2 and 3 together?

Answer: Thank you for your valuable suggestion. In this manuscript, we first treated seven-day-old BALB/c suckling mice with PBS nasal drops, and we wanted to compare the mRNA levels of ROR- γ t and Foxp3 in the lung and intestinal tissues with those of the normal control group, and to test whether there was any effect of PBS on the mRNA levels of ROR- γ t and Foxp3 in the lung and intestinal tissues of seven-day-old BALB/c suckling mice. Our aim was to first demonstrate that the effect of PBS on seven-day-old BALB/c suckling mice was similar to that of normal controls, and that the results were not statistically significant, and then PBS could be used as a negative control before we next began to investigate the effect of RSV infection on seven-day-old BALB/c suckling mice. We hope our explanations can answer your comment.

Comment #2

In the manuscript, the authors propose primer sequences including 16S total length, 16S V3 V4, 16S V4.16S V4 V5, have you all tested them?

Answer: Thank you for your valuable suggestion. We are very grateful to the reviewer for reviewing the paper so carefully. We totally agree with the point raised by the reviewer. Based on the reviewer's comments, we have checked throughout the manuscript. Thanks for your comments to make our manuscript more accurate. The 16S rRNA sequences have both highly variable regions (V regions, which differ between species) and conserved regions (which are highly similar between species) in an alternating arrangement, and the prokaryotic 16S rRNA sequences contain nine highly variable regions. Specific primers were designed for the variable regions of the 16S rRNA gene, and different tag sequences were used to distinguish different samples, and a high-throughput sequencing platform was utilized to perform high-throughput sequencing of different samples, which in turn led to sequencing data. Without considering the conditions of sequencing platform and sequencing quality, the simulation comparison of six commonly used sequential amplified regions revealed that those in the V3-V4 and V4-V5 regions had the highest accuracy in the identification of each taxonomic level [1]. Therefore, in general, the V3-V4 and V4-V5 regions can be preferred for microbial diversity analysis. However, the effect of diversity analysis will also be affected by the primers, and we explored the differences by combining the site information of the primer sequences (ignoring the differences of individual bases in the primers). In this manuscript we chose to examine the 16S V3-V4 region.

Comment #3

The authors well presented that RSV infection in BALB/c mice caused lung immune imbalance, intestinal immune imbalance and intestinal flora disorders. By establishing a model of RSV-infected BALB/c mice, and doing qPCR, HE and intestinal flora sequencing experiments, the authors revealed that RSV infection not only caused an imbalance of Th17/Treg cells in the lungs of the mice, but also caused an excess of IL-22 to be released from the lungs through the circulation. Excessive IL-22, which binds to the IL-22 receptor on the intestinal surface and induces intestinal RegIII γ over-expression, also leads to the release of excessive IL-22 from the lungs through blood circulation, and the binding of IL-22 to the IL-22 receptor on the intestinal surface, which induces impaired Reg III γ overexpression, impaired development of intestinal Th17/Treg, and altered composition of the gut microbiota. The language may need to be further modified and the logic checked.

Answer: Thanks for your comments. In this submission of the manuscript, the relevant inappropriate language has been further revised and the logic has been further checked and corrected. For example, in the section on "Microbial community analysis", we have carefully revised and corrected the "16S amplicon sequencing data analysis" section (Line 175-190): "After DNA extraction and sequencing, the raw paired-end reads underwent a data curation pipeline, which included the removal of low-quality reads (Qiime2 2020.8). Subsequently, the remaining sequences were assigned to their respective samples based on barcode matches, and the barcode and primer sequences were subsequently trimmed. The sequences underwent denoization using the DADA2 method, and reads were classified using the Silva reference database (version 138). A total of 1,324,704 sequence reads were analyzed, with an average of 36,316 reads per sample (range: 23,256 to 45,176). Alpha and beta diversity were computed using Qiime2 2020.8. Principal Coordinate analyses based on the Bray Curtis distance were conducted to assess beta diversity. Chao1 and Shannon indexes were calculated to characterize alpha diversity. Differential analysis was performed utilizing the Wilcoxon rank sum test. Spearman's correlations were used to analyze the relationships between bacterial taxa and metabolites. Correlations were considered significant when the adjusted P values were less than 0.05 after correction for the false discovery rate, using the Benjamini-Hochberg procedure. According to the results of 16S rRNA, the metabolic function of samples was predicted by PICRUST2, the differentially abundant pathways and the composition of specific pathways were also obtained". We are open to further discussion. Thank you for your valuable comments.

Comment #4

In the "Experimental design and sample collection" section of the article, the experimental grouping process should be described in detail, it is not very clear how the groups are grouped and how each group is handled.

Answer: Thanks for your comments. We totally agree with the point raised by the reviewer. We have modified those contents in the "Experimental Design and Sample Collection" section of the article (Line 120-124): Seven-day-old suckling mice were randomly divided into three groups of 24 mice each: normal control group, PBS control group and RSV infection group. For normal control group mice were left untreated, and PBS control group mice were subjected to PBS nasal drip at the same dose as the RSV-infected group mice. For the RSV infection group, the mice were infected with 20 μ L of virus suspension by slow nasal drip. The infection cycle lasted one week, with RSV infection on day one as the benchmark. The mice were weighed daily, then three suckling mice per group randomly selected suckling mice were dissected to collect their lung tissue, small intestine and rectal segments.

Comment #5

Figure 5 is not clearly annotated. Please revise it.

Answer: Thank you for suggesting a change to this section. We are very grateful to the reviewer for reviewing the paper so carefully. We totally agree with the point raised by the reviewer. We have modified the figure note as follows: RSV infection did not cause significant changes in the abundance and diversity of the intestinal flora of suckling mice, but the species composition of the intestinal flora changed. We hope our explanations can answer your comment (Line 602-603).

Comment #6

It makes sense for the authors to write this article, but there is a lack of mechanistic validation, so there should be provide elucidation about the existing limitations of this work and how to follow it up.

Answer: Thank you for your valuable suggestion. The limitations of this work and how to follow them are explained as follows: In this experiment, we chose to collect rectal tissues encased in the feces of lactating mice due to the inconvenience of collecting fecal samples from lactating mice. However, due to the different digestive ability of each lactating mouse, the sample size is not easy to control, which may have an impact on the results of the subsequent bacterial flora assay. We are aware of the limitations of this method and will optimize the experimental design to obtain more accurate results in subsequent studies. Thank you for your valuable comments and suggestions. In order to further understand the effects of bacterial alteration on suckling rats, we found that tryptophan metabolism may have an effect on immunity in the organism from the flora metabolic pathway. Tryptophan is an essential amino acid that the organism cannot synthesize on its own and can only be obtained from food. Studies have shown that tryptophan has anti-inflammatory effects in mammals, and mice fed a low-tryptophan diet are more prone to inflammatory responses. Tryptophan can be metabolized by intestinal flora such as *Escherichia coli*, *Mycobacterium anthropophilum*, and *Aspergillus*, and its metabolites such as indole, 5-hydroxytryptophan, and tyrosine have an important effect on the organism. It has been shown that influenza virus causes an increase in IDO enzyme, which breaks down tryptophan into tyrosine and suppresses immunity [2]. In addition, tryptophan metabolites can regulate IL-22

production through AhR, which plays a key role in immune homeostasis [3]. Subsequently, we will nearly optimize the bacterial colony analysis test and supplemental feeding of tryptophan to lactating suckling mice in vitro to explore its possible role in the body.

Minor questions:

Comment #1

The fonts of some pictures in the article are not very beautiful, such as Figure 1C. Please adjust the layout of the diagram.

Answer: Thank you for your valuable comments. According to the suggestion of the reviewer, in this submission of the manuscript, the layout of Figure 1C has been adjusted to make it aesthetically pleasing. Please check. Thanks.

Comment #2

There are a few punctuation omissions in the article, for example, the page 1 correspondence section, so please double-check the calibration.

Answer. Thanks very much. Some punctuation omissions in the article have been double-checked for calibration in this submission of the manuscript. we hope you will find it in the manuscript line 18-19.

Comment #3

Where the text states that there has been previous research, please add the appropriate literature cited.

Answer: Thank you for your suggestions. We have added the corresponding cited references in the revised manuscript. In response to your suggestion, we went back to the original text and found that it is not that we did not include the appropriate literature, but that we derived relationships based on the previous research, and for the partial revision here, please see line 61-65.

Comment #4

Wouldn't this phrase "the first several of which" be better expressed in a different way?

Answer: Thanks very much. We have changed "the first of which" to "rank among the top of", as detailed in the revised manuscript line 299-300.

Comment #5

**p< 0.01 preceded by *, proposed to be deleted

Answer: Thanks very much. For all the sections containing "***p< 0.01, *p< 0.05" in the text, the "*" in front of them has been deleted, and they have been changed to "P" format with reference to the recent articles published in the journal. We hope you will find them in the manuscript.

Comment #6

Acknowledgement section is a little inappropriate. This part should make a revision.

Answer: Thanks for your valuable comments. We have modified those contents in the Acknowledgement section (Line 425-430): "This study was supported by the Natural National Science Foundation of China (No. 81974306, and No. 82302568), Major Project of Natural Science Research of Anhui Education Department (No. KJ2019ZD23), Research Fund of Anhui Institute of translational medicine (No. 2017zhyx26, 2021zhyx-B07), National first-class undergraduate program construction point (Biotechnology), Projects of Education Quality Engineering of Anhui Province (No. 2021jcxkpy013). It is also available in the manuscript of this submission".

Comment #7

The keywords should be separated by commas. Please revise it.

Answer: Thanks for your comments. Changes have been made to address the use of punctuation between keywords in the text by separating them with commas, please review the revised manuscript as submitted. Please check. Thanks.

Reviewer #2

Comments to the Author (blind)

Figure 1.

While the authors claim success in establishing an RSV infection model in neonatal mice, it is essential to acknowledge Harker et al.'s previous findings on exacerbated disease upon reinfection also in neonatal mice (<https://doi.org/10.1128/jvi.02620-13>).

Answer: Thanks for your comments. As shown in Figure 1C, For the previous study by Harker et al, they found that the disease was exacerbated upon re-infection in neonatal mice. We infected the suckling mice once, while Harker et al. also performed a second infection eight weeks after the first infection of the suckling mice, and different treatments of the suckling mice may have different results, and the results of our experiments by Curcuma are not contradictory to the previous study by Harker et al.

Figure 2.

The claim of successfully establishing a negative control group is unclear; the subsequent experimental analysis predominantly compared the normal control group with the infected group. However, for a more meaningful comparison, it would be more logical to assess differences between the RSV-infected group and the PBS control group. Clarifying this aspect would enhance the clarity and validity of the experimental design and analysis.

Answer: Thanks for your comments. As shown in Figure 2, the mRNA levels of ROR- γ t and Foxp3 in lung tissues of the PBS control group were not significantly different from those of the normal group. After that, RSV nasal drip infection of suckling mice caused a series of changes in the experimental results, can be considered to be caused by RSV infection, RSV-infected group is a virus-positive control group, in contrast to the PBS-treated uninfected RSV group is a negative control group, the mRNA levels of ROR- γ t and Foxp3 in the lung tissues of the PBS control group were not significantly different from those of the normal group, so the paper said that the negative control group was successfully established. Subsequent experimental analyses in this paper were mainly conducted between the normal control group and the infected group, and we believe that the reviewer's suggestion is more logical, but we did analyze the experiments and the results of the PBS control group and the normal control group first, and we found that there were no significant differences between the two, and that is why the experiments and results were analyzed using between the normal control group and the infected group. We hope this answer addresses your concerns.

Figure 3.

When detailing the kinetics of ROR- γ t and Foxp3 expression in lung and intestinal tissues, it is advisable to specify the fold increase compared to the control rather than relying on statistical significance (P value) for clarity in interpretation. This approach provides a more direct and understandable representation of the changes in gene expression over time.

Answer: Thank you for your valuable comments. According to your suggestion, we believe that this approach does provide a more direct and understandable characterization of gene expression over time, and have revised it in this submission, as detailed in this submission.(Line 237-243): As shown in Fig. 3A, the F protein mRNA expression of RSV in the lungs and the intestine was not detected in the normal control group, but on the third day after the virus infection, the greatest amount of expression was observed in the lungs of the RSV infection group ($P < 0.01$), 1-fold increase compared to the first day after infection, and then began to decline. However, no virus was detected in the intestinal tissue of the infected group, indicating that RSV did not directly infect the intestinal tissue. As shown in Fig. 3B and 3C, the mRNA levels of ROR- γ t and Foxp3 in the lung and intestinal tissues were increased in the first 3 days and then decreased, while the mRNA levels of ROR- γ t in the lung tissues were highest on the third day after infection, and the mRNA levels of Foxp3 were the highest on the fourth day after infection, approximately 2-fold and 2.5-fold increase, respectively, compared to normal controls. Please check the updated manuscript. Thanks.

Figure 4.

IL-22 serves dual roles, acting as both a protective and proinflammatory factor, often indicating a mechanism for lung repair. Its function hinges on downstream expression, influenced by chemokines and inflammatory cytokines like IL-1 β , IL-17, and TNF- α . To establish meaningful connections with RegIII γ in intestinal tissue, it is crucial to profile major cytokines originating from the lungs. Examining their profiles in the bloodstream could provide valuable insights, offering hints about potential relationships or correlations. Therefore, more experiments on RT-PCR, ELISA targeting the major cytokines is desired.

Answer: Thanks for your constructive suggestion. We totally agree with the point raised by the reviewer. Regarding your suggestion that examining the characterization of indicators in the blood can provide valuable insights that provide clues to potential relationships or correlations. We think this suggestion you

made is very valuable and we designed this part of the experiment, but the experimental study was conducted on seven-day-old suckling mice, and due to the small size of the mice, their peripheral blood was not collected for the ELISA experiment, which is a limitation of this experiment. In this study, we collected lung and intestinal tissues from each group at different time points, took the supernatant of their grinding fluid, and did ELISA experiments targeting the cytokines IL-10, IL-17, IL-22, IFN- γ , and RegIII γ . The results showed that the excess release of IL-22 from lung tissues may stimulate the expression of Reg III γ in intestinal tissues through the blood circulation, thus affecting the secretion of inflammatory factors by intestinal Th17 and Treg cells. Analysis of the major cytokines from lung-intestinal tissues is essential, and we suggest that a meaningful link to RegIII γ is established in intestinal tissues. For the detection of cytokines in blood, we are in the process of designing experiments in order to collect the blood, and we will re-model the study taking into account the volume of the suckling mice, and we hope to make new discoveries in the near future.

Figure 5. Authors mentions about "indicating that the species composition of the rat intestinal flora changed after RSV infection". Did the author use mice for experimentation as well? The impact of inflammatory cytokines on gut microbiota is established, and daily diet significantly shapes the animal gut microbiome. In disease conditions, such as the inflammation observed in suckling mice, altered dietary patterns may contribute to changes in the gut microbiome. Considering these factors, the observed alterations in the bacteriome profile may not solely be attributed to the RSV-induced cytokine response. To enhance understanding, it is recommended that the authors incorporate another respiratory pathogen model for cross-referencing changes in the bacteriome profile.

Answer: Thank you very much for your valuable suggestions. We have carefully read this sentence in the manuscript, "suggesting that the species composition of the rat intestinal flora changed after RSV infection", and we believe that there is a mistake in our writing. We did not use rats in our study, which was conducted on suckling mice. Please see the corresponding section of the manuscript for the revision of the results in this section. Regarding this suggestion of incorporating another respiratory pathogen model to cross-reference the changes in bacterial group profiles, we think it is a good innovation and very worthwhile for us to carry out the research, and we are about to plan to start the research in this area, and we hope that there will be new research findings in the near future. Thanks again!

[1] Claesson M J, Wang Q, O'sullivan O, et al. Comparison of two next-generation sequencing technologies for resolving highly complex microbiota composition using tandem variable 16S rRNA gene regions. *Nucleic Acids Research*, 2010, 38(22): e200.

[2] Julie MF, Jackelyn MC, Leo KS, et al. Interferon Lambda Upregulates IDO1 Expression in Respiratory Epithelial Cells After Influenza Virus Infection. *Interferon Cytokine Research*, 2015, 35(7): 554-562.

[3] Margaret ML, Jeff EM, Bittoo K, et al. Identification of Cinnabarinic Acid as a Novel Endogenous Aryl Hydrocarbon Receptor Ligand That Drives IL-22 Production. *PLoS One*, 2014, 9(2): e87877.

The following questions have been answered in response to recent submissions to *Spectrum of Microbiology*:

1. The Importance paragraph is currently missing from the Manuscript Text File. The Importance section (150 words or shorter) is a nontechnical explanation of the significance of the study to the field. It should be inserted immediately after the Abstract in the text file (i.e., not just in the eJP submission form). Please add this to your text file.

Answer: Thank you very much for your valuable suggestions. In response to your comment that "the paragraph of importance is currently missing from the text file of the manuscript", we have inserted a section on importance of 150 words or less immediately after the abstract in the text file, as detailed in the latest submission of the manuscript (Line 34-43).

2. Please list the figure legends at the end of the manuscript, after the references section.

Answer: Thank you for suggesting a change to this section. We have moved the figure legend to the end of the manuscript after the references section, see lines 577-623 of the latest submission of the manuscript.

Re: Spectrum03283-23R1 (The imbalance of pulmonary Th17/Treg cells in BALB/c suckling mice infected with respiratory syncytial virus mediated intestinal immune damage and gut microbiota changes)

Dear Dr. Shenghai Huang:

Your manuscript has been accepted, and I am forwarding it to the ASM production staff for publication. Your paper will first be checked to make sure all elements meet the technical requirements. ASM staff will contact you if anything needs to be revised before copyediting and production can begin. Otherwise, you will be notified when your proofs are ready to be viewed.

Sincerely,
Dhammika Navarathna
Editor
Microbiology Spectrum

Reviewer #2 (Comments for the Author):

Dear Authors, Thanks for accepting suggestions.